# Acute Effects of a Maximal Cardiopulmonary Exercise Test on Cardiac Hemodynamic and Cerebrovascular Response and Their Relationship with Cognitive Performance in Individuals with Type 2 Diabetes

**DOI:** 10.3390/ijerph20085552

**Published:** 2023-04-18

**Authors:** Florent Besnier, Christine Gagnon, Meghann Monnet, Olivier Dupuy, Anil Nigam, Martin Juneau, Louis Bherer, Mathieu Gayda

**Affiliations:** 1Research Centre and Centre ÉPIC, Montreal Heart Institute, Montreal, QC H1T 1N6, Canada; 2Department of Medicine, Faculty of Medicine, Université de Montreal, Montreal, QC H3T 1J4, Canada; 3Laboratory MOVE (UR 20296), Faculty of Sport Sciences, Université de Poitiers, 86073 Poitiers, France; 4School of Kinesiology and Physical Activity Sciences (EKSAP), Faculty of Medicine, Université de Montreal, Montreal, QC H3T 1J4, Canada; 5Research Centre, Institut Universitaire de Gériatrie de Montréal, Montreal, QC H3W 1W5, Canada

**Keywords:** near infrared spectroscopy, brain, hemodynamics, cognition, type 2 diabetes mellitus, cardiorespiratory fitness

## Abstract

Cardiovascular and cerebrovascular diseases are prevalent in individuals with type 2 diabetes (T2D). Among people with T2D aged over 70 years, up to 45% might have cognitive dysfunction. Cardiorespiratory fitness ($\dot{V}$O_2_max) correlates with cognitive performances in healthy younger and older adults, and individuals with cardiovascular diseases (CVD). The relationship between cognitive performances, $\dot{V}$O_2_max, cardiac output and cerebral oxygenation/perfusion responses during exercise has not been studied in patients with T2D. Studying cardiac hemodynamics and cerebrovascular responses during a maximal cardiopulmonary exercise test (CPET) and during the recovery phase, as well as studying their relationship with cognitive performances could be useful to detect patients at greater risk of future cognitive impairment. Purposes: (1) to compare cerebral oxygenation/perfusion during a CPET and during its post-exercise period (recovery); (2) to compare cognitive performances in patients with T2D to those in healthy controls; and (3) to examine if $\dot{V}$O_2_max, maximal cardiac output and cerebral oxygenation/perfusion are associated with cognitive function in individuals with T2D and healthy controls. Nineteen patients with T2D (61.9 ± 7 years old) and 22 healthy controls (HC) (61.8 ± 10 years old) were evaluated on the following: a CPET test with impedance cardiography and cerebral oxygenation/perfusion using a near-infrared spectroscopy. Prior to the CPET, the cognitive performance assessment was performed, targeting: short-term and working memory, processing speed, executive functions, and long-term verbal memory. Patients with T2D had lower $\dot{V}$O_2_max values compared to HC (34.5 ± 5.6 vs. 46.4 ± 7.6 mL/kg fat free mass/min; *p* < 0.001). Compared to HC, patients with T2D showed lower maximal cardiac index (6.27 ± 2.09 vs. 8.70 ± 1.09 L/min/m^2^, *p* < 0.05) and higher values of systemic vascular resistance index (826.21 ± 308.21 vs. 583.35 ± 90.36 Dyn·s/cm^5^·m^2^) and systolic blood pressure at maximal exercise (204.94 ± 26.21 vs. 183.61 ± 19.09 mmHg, *p* = 0.005). Cerebral HHb during the 1st and 2nd min of recovery was significantly higher in HC compared to T2D (*p* < 0.05). Executive functions performance (Z score) was significantly lower in patients with T2D compared to HC (−0.18 ± 0.7 vs. −0.40 ± 0.60, *p* = 0.016). Processing speed, working and verbal memory performances were similar in both groups. Brain tHb during exercise and recovery (−0.50, −0.68, *p* < 0.05), and O_2_Hb during recovery (−0.68, *p* < 0.05) only negatively correlated with executive functions performance in patients with T2D (lower tHb values associated with longer response times, indicating a lower performance). In addition to reduced $\dot{V}$O_2_max, cardiac index and elevated vascular resistance, patients with T2D showed reduced cerebral hemoglobin (O_2_Hb and HHb) during early recovery (0–2 min) after the CPET, and lower performances in executive functions compared to healthy controls. Cerebrovascular responses to the CPET and during the recovery phase could be a biological marker of cognitive impairment in T2D.

## 1. Introduction

According to the International Diabetes Federation, 537 million adults worldwide had diabetes in 2021 [1]. In Canada, 11 million people live with prediabetes or diabetes. This number is expected to increase to 13.9 million (33% of Canadians) by 2026 [2]. Micro and macrovascular complications are prevalent in diabetic patients [3], thus increasing the risk of coronary heart disease and stroke, even when glycaemia is well controlled by medication [4]. The frequency and severity of complications are closely related to diabetes duration and to the patient’s age [5,6]. Among people older than 60 years with type 2 diabetes (T2D), up to 20% of them might develop dementia [7]. Brain insulin resistance [8], neuro-inflammatory disorders [9], oxidative stress [7], sympathovagal imbalance [10] and impaired dynamic cerebral autoregulation [11] have been proposed to explain cerebrovascular disease and cognitive dysfunction in T2D individuals [6,7,12]. More specifically, executive function [13], in particular cognitive flexibility and verbal fluency [14], and verbal episodic memory [14] are decreased in individuals with T2D with Cohen’s d ranged from −0.22 to −0.51 compared to healthy controls [12,14,15].

Regular physical activity (PA) and exercise training are the cornerstones of managing patients with T2D [16]. Regular PA enhances glycemic control, general health, cardiorespiratory fitness, and reduces cardiovascular risk factors such as obesity [16]. Regular PA also has psychological and cognitive benefits for people with T2D [16,17]. Cardiorespiratory fitness ($\dot{V}$O_2_max) correlates with cognitive performance in healthy young subjects, older adults [18], and individuals with cardiovascular diseases [19,20]. This suggests that cardiorespiratory fitness plays a crucial role in maintaining and improving cognitive functions. $\dot{V}$O_2_max could influence cerebral blood flow and cerebral oxygenation during exercise in healthy or in obese individuals [21,22]. In CHD patients, we showed that cognitive performance was related to $\dot{V}$O_2_max, maximal cardiac output and a reduced cerebral oxygenation/perfusion response during exercise [19]. As well, we showed the protective effects of a higher $\dot{V}$O_2_max on cognition in obese patients vs. non-obese peers [23]. However, to date, the relationship between $\dot{V}$O_2_max, cardiac/cerebral hemodynamic and cognitive function has been poorly studied in patients with T2D.

Patients with T2D have impaired $\dot{V}$O_2_max and cardiac function, as well as reduced cerebral vasodilatory capacity, cerebral blood flow and cerebral oxygenation during exercise [24,25,26]. Nevertheless, the relationship between these physiological parameters during exercise and cognitive performance is poorly studied in patients with T2D. Studying cardiac hemodynamic and cerebrovascular responses during a maximal cardiopulmonary exercise test (CPET) and during its recovery phase, as well as their relationship with cognition could be useful to detect patients at greater risk of cognitive decline. To date, this relationship has never been studied in individuals with T2D despite their higher risk of cerebral, cognitive, and cardiovascular dysfunction.

The objectives of this study: (1) to compare cerebral oxygenation/perfusion during a CPET and during its post-exercise period (recovery); (2) to compare cognitive performance in patients with T2D to those in healthy controls; and (3) to examine if $\dot{V}$O_2_max, maximal cardiac output and cerebral oxygenation/perfusion will be associated with cognitive function in T2D and HC subjects.

Our main hypotheses: (1) Cardiorespiratory fitness, cerebral oxygenation and cognitive function would be reduced in T2D vs. HC. (2) Cardiorespiratory fitness and cerebral oxygenation would be correlated with cognition in T2D and HC.

## 2. Material and Methods

### 2.1. Participants

Nineteen individuals with type 2 diabetes mellitus (T2D) and 22 age-matched healthy controls (HC) were evaluated at the Montreal Heart Institute’s Preventive Medicine and Physical Activity Centre (ÉPIC). Inclusion criteria for T2D patients: aged >18 years, previously diagnosed diabetes defined by fasting glycemia ≥7.0 mmol/L and HbA1c >0.06. For HC, inclusion criteria: >18 years, no cardiovascular risk factors, or heart disease. For both groups, exclusion criteria: insulin therapy, previous cardiovascular diseases, malignant arrhythmias during exercise testing, restriction to cardiopulmonary exercise testing or severe intolerance to exercise. The study protocol (ClinicalTrials.gov: NCT01906957) was approved by the Montreal Heart Institute’s Research Ethics Board. Written informed consent was obtained from all participants prior to inclusion. The investigation was conducted in accordance with the principles outlined in the Declaration of Helsinki. The flow chart of the study is indicated in Figure 1.

### 2.2. Measurements

Firstly, all subjects were met by a medical practitioner for a physical and medical assessment and to check study eligibility. Then, a blood sample was taken by a nurse and the body composition analysis was collected using a bioimpedance scale (Tanita, model BC418, Japan) by a research assistant. Next, subjects performed a cognitive test assessment, followed by a maximal cardiopulmonary exercise test with gas exchange analysis (CPET). During the CPET, cerebral oxygenation/perfusion (measured by near-infrared spectroscopy) and cardiac hemodynamic responses (impedance cardiography) were measured continuously.

### 2.3. Maximal Cardiopulmonary Exercise Testing (CPET)

Every subject realized a maximal incremental test (CPET) on a bicycle ergometer (Ergoline 800S, Bitz, Germany) using an incremental personalized protocol (ten to twenty watts per min stages), as previously published [19,20,22]; see indicated references for details.

Continuous electrocardiogram monitoring (Marquette, case 12, St. Louis, MO, USA), rating of perceived exertion (Borg Scale, 6–20) and manual blood pressure using a sphygmomanometer (Welch Allyn Inc., Skaneateles Falls, USA) every 2 min were monitored throughout the test [19,20,22]. Subjects used a facemask covering their mouth and nose which was attached to the flow module and to the gas sampling line. The calibration of the flow module was accomplished by introducing a calibrated volume of air at several flow rates with a 3-L pump. Gas analyzers were calibrated before each test using a standard certified commercial gas preparation (O_2_: 16%; CO_2_: 5%). Cardiopulmonary variables [(minute ventilation ($\dot{V}$_E_: L/min), oxygen uptake ($\dot{V}$O_2_: mL/min) and carbon dioxide production ($\dot{V}$CO_2_: mL/min)] were recorded (Oxycon Pro; CareFusion, Jaeger, Germany) during the CPET phase (3-min rest and warmup, effort, and 5-min recovery) according to a previously published methodology [19,20,22]. All subjects were encouraged to provide a maximal effort. The participant reached maximal effort when one of the following four criteria were met: (1) the attainment of the primary maximal criteria: a levelling off of oxygen uptake (<150 mL/min) despite increased power; (2) a respiratory exchange ratio >1.10; (3) an inability to maintain 60 rpm; or (4) patient exhaustion due to fatigue or other clinical symptoms (dyspnea, ECG and/or blood pressure abnormalities). The 1st ventilatory threshold (VT1) was determined using a combination of the V-slope, ventilatory equivalents, and end-tidal oxygen pressure methods. The highest value of O_2_ uptake was defined as the maximal value ($\dot{V}$O_2_max). The presence of a $\dot{V}$O_2_ plateau was defined by an increase of $\dot{V}$O_2_ values <150 mL/min during the last 30 s of the exercise phase. The maximal power output (PPO) was defined as the power (Watts) of the last fully completed one minute stage [19,20,22].

### 2.4. Cardiac Hemodynamics during CPET

Cardiac hemodynamics were measured continuously through the CPET using a non-invasive impedance cardiography device (PhysioFlow, Enduro model, Manatec, France), previously found to be valid, accurate, and reproducible at rest and during exercise in healthy subjects [27]. Data were averaged every 15 consecutive heartbeats for cardiac index (CI: in L/min/m^2^), stroke volume index (SVi: in mL/m^2^), heart rate (in beats/min), stroke volume index (mL/m^2^), left cardiac work index (LCWi: in kg·m/m^2^) and systemic vascular resistance index (SVRi: in dynes/s/cm^5^/m^2^) [19,20,22].

### 2.5. Cerebral Oxygenation/Perfusion (NIRS)

Cerebral oxygenation/perfusion was measured using a near-infrared spectroscopy (NIRS) system (Oxymon Mk III, Artinis Medical, Netherlands) during the CPET (during 3 min rest and warm-up, exercise phase and 5-min recovery: 2 min active recovery pedaling at low workload (20 W) and 3 min of passive recovery). Optodes were placed on the left prefrontal cortical area between Fp1 and Fp3, according to the modified international EEG 10–20 system [19,22]. Relative concentration changes (µmol/L) were measured from resting baseline of oxyhemoglobin (O_2_Hb), deoxyhemoglobin (HHb), and total hemoglobin (tHb). The baseline period for exercise was set at the end of the 3-min resting period, defined as 0 µmol/L [19,22]. Data were displayed in real time and stored on disk for off-line analysis. Raw NIRS signals were filtered via the oxysoft/DAQ software (Artinis Medical, Netherlands) using a running average function with a filter width of 1 [19,22]. Thereafter, NIRS signals were exported into excel files with the oxysoft/DAQ software at 0.2 Hz for statistical treatment. During the exercise test, optodes were secured with a tensor bandage wrapped around the forehead, a neoprene pad was placed between the skin and the optodes’ plastic holder and ambient room light was reduced. The room temperature and relative humidity were constant in the laboratory (21 °C, 25%). To correct for scattering of photons in the tissue, a differential path-length factor of 5.93 was used for the calculation of absolute concentration changes with an interoptode distance of 45 mm [19,22]. Data were sampled at 10 Hz during the rest period (3 min), the exercise phase and the 5-min recovery period [19,22]. The mean values for O_2_Hb, HHb and tHb were analysed at rest, and at 25%, 50%, 75% 100% of maximal power output and for each minute of the recovery period. The area under the curve for O_2_Hb, HHb and tHb (exercise and 5 min recovery period) were then calculated.

### 2.6. Blood Samples Analyses and Neuropsychological Evaluation

A fasting blood draw measuring lipids (total, HDL, and LDL-cholesterols and triglycerides), glycemia and insulin levels by the Montreal Heart Institute’s medical biochemistry laboratory as previously published [19,20,22].

Neuropsychological test battery

Short-term memory, working memory, long-term verbal memory, executive functions and processing speed, were evaluated [19,20,23,28]. In addition, the Mini Mental state examination (MMSE), a measure of global cognition [29], was also recorded. Finally, a self-reported questionnaire measuring depressive symptoms, the Geriatric Depression Scale (GDS) [30], was administered. In order to test their short-term and working memory, participants were asked to repeat a series of numbers given orally by the examiner (one per second) in both the forward and the backward order. Then, participants completed the Digit Symbol Substitution Test (DSST) [31], which required them to match symbols to numbers (1–9). Participants had 120 s to draw as many associations as possible. The DSST evaluates processing speed. In part A of the Trail Making Test (TMT), participants had to quickly link numbers (from 1 to 25) with straight lines. In part B, they had to alternate between linking letters in alphabetical order and numbers in ascending order (1-A-2-B-3-C, etc.) [32].

Part A of the TMT targets processing speed, whereas part B targets switching abilities. The D-KEFS Color-Word Interference Stroop Test includes 4 conditions [33]. Participants had to identify the different colour of each rectangle present on a sheet of paper (the naming condition). In the reading condition, participants had to read aloud words printed in black that represent colours such as “red” or “blue”. In the inhibition condition, the meaning of the word was printed in a colour that did not match that word (for example, the word “red” is printed in green). Participants were instructed to identify the colour of the ink (green) without reading the word. In the switching condition, they had to switch to reading the words while ignoring the actual ink color when the words were surrounded by a square. Participants were instructed to responded as quickly and correctly as possible. The first two conditions of the Stroop assess processing speed, whereas the third and fourth assess executive functions, i.e., inhibition and switching abilities. Finally, participants had to learn and remember a list of 15 words that was read to them on 5 consecutive trials for the Rey Auditory Verbal Learning Test (RAVLT, episodic verbal memory) [28]. During this task, the immediate recall was evaluated after an interfering list, while the delayed recall was measured after a 30-min delay [19,20,23].

### 2.7. Composite Scores for Cognition

Standardized Z-scores (Z-score = (value–mean value of all the subjects)/standard deviation) were constructed for all cognitive scores. Then, four composite cognitive scores were calculated using raw Z-scores as follows: (1) Working memory = ((DS forward + DS backward scores)/2); (2) Processing speed = ((DSST + TMT A + Stroop 1 + Stroop 2)/4); (3) Executive functions = ((Trail B + Stroop 3 + Stroop 4)/3); and (4) Verbal memory/episodic memory (immediate recall + delayed recall + total words recalled during the 5 learning trials from the RAVLT test/3) [20]. The internal consistency of all the measures that made up a composite score was examined using Cronbach’s alphas (α), with a Cronbach’s > 0.7 being considered as adequate. The composite scores’ internal consistency was good for working memory with a Cronbach’s alpha (α) of 0.717; for processing speed, α = 0.761; and for executive functions, α = 0.911; but not for verbal memory, α = 0.340.

### 2.8. Statistical Analysis

Normal Gaussian distribution of the data was verified graphically and by Shapiro–Wilk tests. Data were compared between the two groups (HC and T2D) using a Student *t*-test (for normally distributed data) or a Mann-Whitney test (for abnormal distribution). For categorical variables, percentages were compared between groups with a chi-square test. All variables were presented as means ± standard deviation (SD) as appropriate for continuous ones, and in numbers and percentage for categorical. Relationships between cardiorespiratory fitness, cardiac impedance, blood parameters, NIRS (highest values and AUC) and the four main cognitive Z-scores were assessed with a Pearson correlation coefficient (r). Bonferroni-corrected post hoc tests were used if needed to localize group differences. All statistical tests were two-sided and conducted at a 0.05 significance level. Statistical analyses were performed with the use of Stata SE 15.1 (StataCorp LP, College Station, TX, USA) and GraphPad Prism 9 (GraphPad Software, Inc., La Jolla, CA, USA).

## 3. Results

### 3.1. Clinical Characteristics

Twenty-three HC and 19 T2D subjects preformed the examinations. Baseline characteristics of HC and T2D are shown in Table 1. T2D took optimal pharmacological treatment. Age was statistically identical in our groups. Both groups were composed in majority of male participants and the male/female ratio was similar in each group (*p* = 0.483). As expected, BMI values and cardiovascular risk factors were higher in T2D than in healthy subjects (Table 1).

### 3.2. Cardiopulmonary Exercise Test and Cardiac Hemodynamic Parameters

The CPET and cardiac hemodynamics data are presented in Table 2. T2D patients had lower $\dot{V}$O_2_max values even expressed in mL/min/kgFFM compared to HC (*p* < 0.001). Maximal cardiac index, and left cardiac work index were both lower in T2D patients compared to HC (*p* < 0.05). Systemic vascular resistance index and systolic blood pressure at maximal exercise were higher in T2D (*p* = 0.005).

### 3.3. Cerebral Oxygenation/Perfusion during Exercise and Recovery

Left prefrontal NIRS variables during exercise were compared between groups at rest, and at 25%, 50%, 75% and 100% of maximal power output (Figure 2a). During exercise, no differences were found for O_2_Hb, HHb, tHb and AUC at either exercise intensity between HC and T2D patients. During recovery, O_2_ Hb at the beginning of the recovery (0) and HHb during the 1st and 2nd minutes of recovery were significantly higher in HC compared to T2D (*p* < 0.05) (Figure 2b). Area under the curve (AUC) values were computed and compared between groups (Table 3).

Composite scores for executive functions were significantly lower in T2D patients compared to HC (*p* = 0.016). Processing speed, working and verbal memory composite scores were similar in both groups. Results of cognitive tests and of cognitive composite Z-scores are presented in Table 4.

### 3.4. Relationship between Cognitive Performance, Cardiorespiratory Fitness and Cerebral Hemodynamics

The relationship between cognitive performance (Z-scores in each domain), cardiorespiratory fitness, cardiac impedance and cerebral oxygenation (AUC during exercise and recovery) was evaluated separately in T2D and HC (Table 5). AUC for tHb during exercise and recovery, and AUC O_2_Hb during recovery only negatively correlated with executive functions performance (*p* < 0.05) in T2D patients, such that lower AUC tHb values were associated with slower response times. For HC, but not for T2D, $\dot{V}$O_2_max (mL/min/kgFFM), cardiac index, AUC tHb, AUC O_2_Hb during exercise and recovery were all negatively correlated with executive function (*p* < 0.05); lower AUC tHb values were associated with slower response times. Processing speed only correlated negatively with AUC tHb and O_2_Hb during exercise in HC, with lower values being correlated with slower response times. None of the parameters were related to processing speed in T2D patients, and none of them correlated with verbal memory in either group.

## 4. Discussion

The originality of our study was to simultaneously assess cardiopulmonary function, cardiac and cerebral hemodynamics during exercise and its recovery, in addition to performing a complete cognitive assessment. The main results can be summarized as follows: (1) in addition to a reduced $\dot{V}$O_2_max, cardiac hemodynamic (CI, SVi) and elevated vascular resistance (SVRi), T2D patients showed reduced cerebral hemoglobin values (O_2_Hb and HHb) during early recovery (0–2 min) after maximal exercise; (2) T2D patients had lower executive functions performance at rest (composite score) compared to HC; (3) executive functions performance correlated with cerebral total hemoglobin values during exercise and recovery, as well as with O_2_Hb during recovery in T2D patients (lower values were related to slower response times).

### 4.1. Cardiopulmonary Function and Cardiac Hemodynamic at Maximal Exercise

We showed that T2D patients have reduced normalized $\dot{V}$O_2_max (by FFM) as compared to healthy controls, as well as reduced maximal ventilation, indicating a reduced ventilatory convection. The impaired pulmonary function (expiratory reserve volume and/or functional reserve capacity) often observed in obese patients (very prevalent in our T2D group) can be the cause of the increased respiratory load and cost [34]. In addition, maximal cardiac index was also reduced, indicating an impaired cardiac function in T2D patients [25]. This reduction was done through lower maximal heart rate and stroke volume index, but also through higher end-systolic-volume-index and systemic-vascular-resistance-index. This indicates a higher exercise cardiac afterload. Our results are in agreement with previous studies, showing systolic and/or diastolic dysfunctions during exercise in T2D in addition to less compliant systemic vessels [25,35,36]. However, we did not show any difference in O_2_ extraction (C (a-$\bar{v}$ O_2_)) by muscles between T2D and HC subjects. Nevertheless, muscle blood flow and mitochondrial dysfunction have been observed in T2D and could impact O_2_ extraction in other studies [35].

### 4.2. Cerebral Hemodynamic during Exercise and Recovery

We showed that during CPET, cerebral hemodynamics (O_2_Hb, tHb, HHb) were similar between T2D patients and healthy controls. The main regulators of cerebral hemodynamics during exercise include partial pressure of carbon dioxide (PaCO_2_), arterial blood pressure and cardiac output [37]. In our study, patients with T2D had higher maximal systolic blood pressure values, lower maximal cardiac output and lower maximal ventilation, as well as higher PETCO_2_ at maximal exercise compared to healthy controls. The study by Kim et al. showed reduced PaCO_2_ (−0.8 mmHg) cerebral blood flow (≈−25%) and oxygenation (−2.5%) (measured via catheters) in T2D patients compared to controls. In addition, their study showed a reduced cardiac output, stroke volume and vascular conductance during exercise in these individuals [25]. Another study in patients with T1D (with inadequate glycaemic control) showed reduced brain O_2_Hb, tHb and HHB (−1.25; −6; −6.5 µM at 90–100% $\dot{V}$O_2_max) during exercise compared to controls [38]. In the present study we did not witness beneficial aerobic fitness effects on cerebral hemodynamics (O_2_Hb, tHb and HHB) in T2D, in agreement with our previous study in obese patients [22]. Age, gender, genetic factors, insulin resistance, other cardiovascular risk factors, and perhaps drug treatments are all factors that can influence arterial vasomotricity and the relationship between cardiorespiratory fitness and cerebral circulation [39]. Other studies showed that fitter young adults can have higher O_2_Hb, tHb and HHb values compared to less fit peers [21,40]. However, during recovery, we showed that our T2D patients showed reduced O_2_Hb (at T0) and HHb (T0-120 s) comparatively to healthy controls. This indicates that T2D patients have reduced cerebral O_2_ extraction during recovery, as indicated by lower HHb value vs. healthy controls [41]. In T2D, it has been suggested that glycation of hemoglobin can reduce the kinetics of O_2_Hb release. Another potential mechanism is the reduced glucose metabolism and O_2_ use in neurons (neurovascular coupling) [38]. These results differ from previous studies with obese patients comparing post-exercise cerebral hemodynamics (no existing studies available in T2D). We and others found similar cerebral hemodynamics (O_2_Hb, tHb and HHb) during exercise recovery in obese participants [19,42,43]. We also demonstrated that cerebral oxygen extraction (higher HHb levels) might be improved with interval training in obese patients [41]. The experimental setting used in the present study does not conclusively demonstrate the mechanism leading to higher HHb changes in healthy adults compared to T2D participants. However, it is possible that in T2D individuals the greater impairment of vasodilatory capacity in the cerebral vasculature (reduced cerebral CO_2_ responsiveness), the greater reduction in flow-mediated dilatation (attributed to reduced nitric oxide bioavailability), and the presence of hyperventilation during recovery (and corresponding hypoperfusion) were limiting the capacity of the vascular wall to accommodate oxygenated blood [38]. Hence, healthy adults that are metabolically more efficient extracted and consumed more oxygen from the existing regional blood volume; this is reflected as higher HHb concentration changes in the control group compared with T2D patients. Brain imaging such as fNIRS may provide a biomarker of cerebrovascular dysfunction in T2D [44]. Further studies are needed to refine and/or confirm this interpretation.

### 4.3. Cognitive Function

We showed that individuals with T2D have reduced executive functions performance (lower composite Z-scores for executive function) as compared to healthy controls. Hyperglycemia-induced vascular complications such as neuropathy can lead to cognitive dysfunction in patients with T2D. In turn, cognitive impairment could interfere with diabetes self-management and reduce adherence to proper treatment. Executive functions, processing speed, attention and memory are the most affected cognitive domains, with performances in individuals with T2D that are 0.3 to 0.4 standard deviations lower compared to healthy controls [12]. There is no study showing a mediating effect of cardiorespiratory fitness on cognitive performance in T2D, but we previously showed that fit obese individuals have better cognitive performances compared to their unfit obese peers, and have similar cognitive performances to those of non-obese controls [23]. Memory performance was associated with body composition changes in previous studies in T2D [45]. Strategies to improve both cardiorespiratory fitness and cognition have been proposed, such as exercise training [46].

### 4.4. Relationship between Cognitive Performance, Cardiorespiratory Fitness and Cerebral Hemodynamics

We demonstrated a relationship between executive function performances, aerobic fitness, cardiac output and exercise-related cerebral hemodynamics (AUC: O_2_Hb, tHb) in the healthy control group. In T2D, only cerebral total haemoglobin (AUC) during exercise was related to executive function performances. In addition, cerebral hemodynamics during recovery (AUC: O_2_Hb, HHb) were also correlated with executive function performances in healthy controls. In T2D, cerebral hemodynamics during recovery (AUC: O_2_Hb, tHb) were correlated with executive function performances. Finally, processing speed performances were related to exercise cerebral hemodynamics (AUC: O_2_Hb, tHb) only in healthy controls. Our results are partly in agreement with the vascular hypothesis, which suggests that lower $\dot{V}$O_2_max, cardiac function and cerebrovascular reserve (measured by NIRS) during exercise may be related to worse cognitive function [19,47]. This relationship seemed to be less constant or absent in the T2D group. In a previous study in obese patients, we showed that cognitive performance (executive function, short-term memory, processing speed) was independently predicted by normalized aerobic fitness ($\dot{V}$O_2_maxFFM) [23]. We also showed in a sample of adults with or without heart disease that aerobic fitness ($\dot{V}$O_2_maxFFM), cardiac output and cerebral hemodynamics (effort and recovery) were related to cognitive performance [19,20].

## 5. Limitations and Perspectives

Our study has limitations, including a low sample of non-diabetic subjects and patients with T2D, including mostly men, recruited in a single institution. Our results may therefore be different in female participants, with or without T2D. Cerebral oxygenation and perfusion was also measured superficially and non-invasively during exercise using NIRS in the left prefrontal cortical area, with one channel. Therefore, our brain hemodynamic results may differ from other more global measurements of brain oxygenation and perfusion (e.g., catheters) or from other brain regions (e.g., fNIRS). We cannot exclude that some medication taken by T2D may affect aerobic fitness. Cognition as well as cardiac and cerebral hemodynamic responses during maximal exercise and recovery should be studied in these two populations (fit and unfit T2D).

## 6. Conclusions

The present study showed that TD2 patients had reduced $\dot{V}$O_2_max, cardiac hemodynamic and elevated vascular resistance, as well as reduced cerebral hemoglobin (O_2_Hb and HHb) during early recovery (0–2 min) after maximal exercise. T2D patients also showed lower cognitive performance in executive function tests compared to healthy controls. Cerebrovascular responses during incremental exercise and recovery could be biological markers of cognitive impairment in T2D. Further studies are needed.

## Figures and Tables

**Figure 1 ijerph-20-05552-f001:**
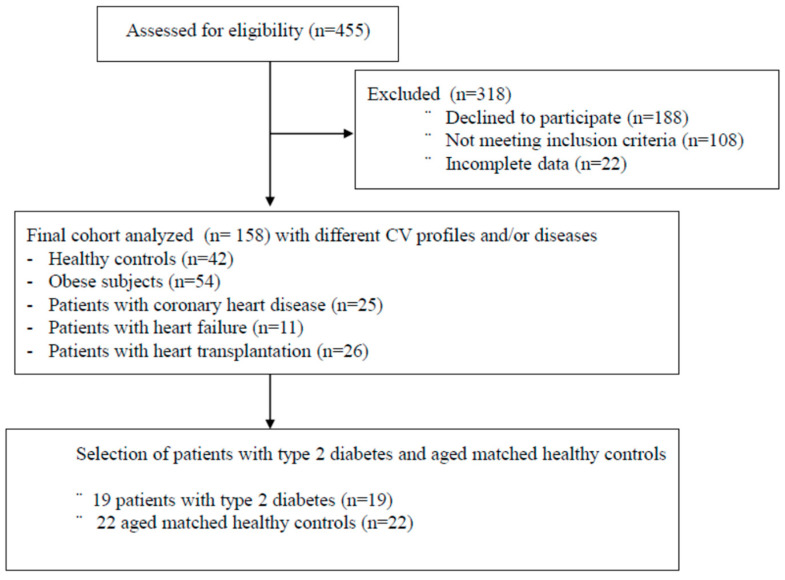
Flowchart of the inclusion of the subjects of the COGNEX study.

**Figure 2 ijerph-20-05552-f002:**
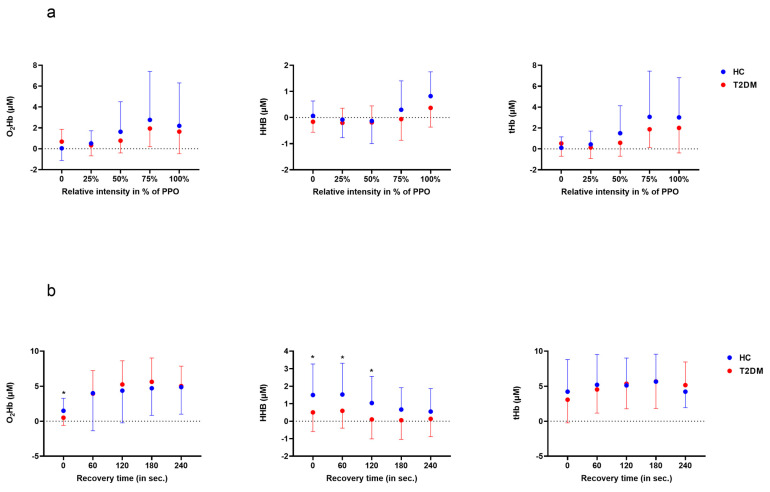
Cerebral oxygenation/perfusion variables (NIRS) during acute exercise (**a**) and its recovery (**b**) in patients with type 2 diabetes (T2D) and healthy controls (HC). PPO: maximal power output. * = *p* < 0.05.

**Table 1 ijerph-20-05552-t001:** Baseline clinical characteristics of healthy controls (HC) and patients with T2D.

	HC *n* = 23			T2D *n* = 19			
	Mean	±	SD	Mean	±	SD	*p* Value
Age	61.83	±	10.54	61.95	±	6.60	0.964
Sex (female, %)	4 (17.4%)			5 (26.3%)			0.483
Height (cm)	171.00	±	8.73	169.79	±	9.96	0.677
body mass (kg)	71.66	±	7.58	90.96	±	14.22	<0.001
BMI (kg/m^2^)	24.51	±	1.93	31.67	±	5.50	<0.001
Waist circumference (cm)	89.43	±	6.40	111.42	±	14.36	<0.001
Fat free mass (kg)	56.83	±	8.69	59.92	±	10.57	0.305
Fat mass (%)	21.00	±	6.24	33.70	±	8.56	<0.001
Fat mass (kg)	14.87	±	3.78	30.84	±	9.35	<0.001
**Medication**							
Beta-blocker	0 (0%)			3 (16%)			
ACE inhibitor	0 (0%)			9 (47%)			
ARBs	0 (0%)			2 (11%)			
Calcium channel blocker	0 (0%)			5 (26%)			
Antidiabetic agent	0 (0%)			9 (47%)			
**Cardiovascular risk factors**							
Smoking (*n*, %)	0 (0%)			3 (15.8%)			0.048
Hypertension (*n*, %)	0 (0%)			12 (63.2%)			<0.001
Dyslipidemias (*n*, %)	2 (8.7%)			15 (78.9%)			<0.001
Overweight: BMI > 25 and ≤ 30 kg·m^2^	8 (34.8%)			7 (36.8%)			0.889
Obesity: BMI > 30 kg·m^2^	0 (0%)			11 (57.9%)			<0.001
**Blood sample**							
Fasting glycemia (mmol/L)	5.10	±	0.54	7.72	±	1.48	<0.001
Total cholesterol (mmol/L)	5.01	±	0.86	4.23	±	1.10	0.018
HDL-C (mmol/L)	1.53	±	0.50	1.12	±	0.26	0.003
LDL-C (mmol/L)	3.08	±	0.68	2.31	±	0.97	0.013
Triglycerides (mmol/L)	0.90	±	0.40	1.90	±	1.00	<0.001
Fasting insulin (pmol/L)	47.30	±	69.07	110.71	±	65.97	0.001

BMI: body mass index.

**Table 2 ijerph-20-05552-t002:** Cardiorespiratory fitness and cardiac impedance between healthy controls (HC) and T2D.

	HC *n* = 23			T2D *n* = 19			
**Cardiorespiratory Fitness**	**Mean**	**±**	**SD**	**Mean**	**±**	**SD**	***p* Value**
Time to exhaustion (sec)	666.00	±	177.20	589.20	±	143.30	0.1467
Power at VT1 (watts)	154.80	±	52.97	107.50	±	31.94	0.0033
$\dot{V}$O_2_ at VT1 (mL/min)	2006.00	±	564.50	1562.00	±	348.30	0.0090
Power increment (w/min)	15.48	±	2.69	13.53	±	2.93	0.0418
$\dot{V}$O_2_max (mL/kg/min)	36.89	±	7.66	22.84	±	4.89	<0.001
$\dot{V}$O_2_max (mL/kg FFM/min)	46.43	±	7.63	34.50	±	5.65	<0.001
$\dot{V}$O_2_max (mL/min)	2666.91	±	682.34	2057.95	±	450.04	0.002
$\dot{V}$O_2_ Plateau (*n* and %)	20 (86%)			16 (68%)			0.1450
R.E.R	1.17	±	0.07	1.16	±	0.08	0.431
$\dot{V}$Emax (L/min)	105	±	39.2	75.5	±	20.9	0.015
PETCO_2_max (mmHg)	34.33	±	4.47	38.03	±	4.53	0.014
HR rest (bpm)	65.13	±	12.26	74.21	±	14.65	0.015
HR max (bpm)	158.96	±	14.20	144.74	±	24.12	0.031
Systolic blood pressure (mmHg)	119.30	±	7.90	127.74	±	14.18	0.028
Diastolic blood pressure (mmHg)	71.91	±	7.32	75.37	±	8.69	0.169
Maximal systolic blood pressure (mmHg)	183.61	±	19.09	204.95	±	26.21	0.004
Maximal diastolic blood pressure (mmHg)	79.13	±	10.49	79.79	±	10.50	0.840
HR at 1 min of recovery (bpm)	139.22	±	16.24	120.58	±	23.90	0.005
Maximal power output (watts)	214.52	±	68.21	149.06	±	43.44	0.002
**Cardiac impedance at maximal exercise**							
Cardiac output (L/min)	15.86	±	2.12	12.82	±	4.16	0.008
Cardiac Index (L/min/m^2^)	8.70	±	1.09	6.27	±	2.09	<0.001
Stroke volume (mL)	101.04	±	11.41	97.39	±	31.57	0.636
Stroke volume index (mL/m^2^)	54.75	±	4.45	47.03	±	15.03	0.043
Left cardiac work index (kg/m/m^2^)	12.77	±	2.10	9.90	±	3.47	0.025
Contractility Index	147.35	±	55.20	106.68	±	45.99	0.024
C(a-$\bar{v}$ O_2_) (mL/100 mL)	16.71	±	3.25	17.12	±	4.48	0.732
Systemic vascular resistance index (Dyn·s/cm^5^·m^2^)	583.35	±	90.36	826.21	±	308.21	0.005
Systemic vascular resistance (Dyn·s/cm^5^)	1071.09	±	157.06	1986.21	±	1297.65	<0.001

VT1: 1st ventilatory threshold; FFM; fat free mass; R.E.R: respiratory exchange ratio; PETCO_2_: partial pressure of end-tidal CO_2_; HR: heart rate; C(a-$\bar{v}$O2): difference in the O2 content of the blood between the arterial blood and the venous blood.

**Table 3 ijerph-20-05552-t003:** Cerebral oxygenation/perfusion AUC during a maximal cardiorespiratory exercise and the post-exercise period in healthy controls (HC) and T2DM.

NIRS at Exercise and during Recovery	HC *n* = 23			T2DM *n* = 19			*p*
AUC exercise tHb	6.57	±	9.79	3.87	±	4.35	0.245
AUC exercise HHb	0.53	±	2.77	−0.34	±	2.30	0.279
AUC exercise O_2_Hb	6.03	±	10.42	4.21	±	3.79	0.444
AUC exercise HbDiff	5.50	±	11.70	4.56	±	4.52	0.318
AUC recovery tHb	18.31	±	11.95	19.70	±	13.66	0.731
AUC recovery HHb	4.26	±	5.59	1.07	±	3.97	0.048
AUC recovery O_2_Hb	16.27	±	15.00	17.60	±	11.18	0.757
AUC recovery HbDiff	13.52	±	20.28	17.56	±	12.75	0.465
AUC: Area Under the Curve							

AUC: Area Under the Curve.

**Table 4 ijerph-20-05552-t004:** Cognitive evaluation in healthy controls (HC) and T2DM.

	HC *n* = 23			T2M *n* = 19			
	Mean	±	SD	Mean	±	SD	*p* Value
MMSE	28.94	±	1.16	28.94	±	1.20	0.902
GDS	2.35	±	2.78	2.35	±	3.67	0.340
Forward	6.83	±	0.92	6.47	±	0.94	0.258
Backward	5.56	±	1.50	5.35	±	1.46	0.688
TMT-A	36.76	±	12.27	38.46	±	12.33	0.685
TMT-B	76.23	±	19.20	83.45	±	31.03	0.415
Stroop 1	29.61	±	5.67	29.77	±	5.20	0.934
Stroop 2	20.91	±	3.37	21.61	±	3.39	0.552
Stroop 3	53.35	±	9.66	63.06	±	13.62	0.021
Stroop 4	56.42	±	13.83	69.38	±	17.96	0.024
Rey first recall	11.94	±	2.41	9.76	±	3.31	0.036
Rey diff	10.67	±	2.72	9.71	±	2.80	0.311
Rey 1–15 total words	52.61	±	9.23	46.41	±	9.75	0.062
DSST	69.72	±	11.42	62.65	±	13.32	0.100
**Cognitive Z score**							
Working memory	0.12	±	0.79	−0.13	±	0.76	0.349
Processing speed	−0.02	±	0.44	−0.06	±	0.53	0.353
Verbal memory	0.27	±	0.80	−0.29	±	0.98	0.069
Executive function	−0.40	±	0.60	0.18	±	0.71	0.016

MMSE: Mini-Mental State Examination; GDS: The Geriatric Depression Scale; TMT: trail making test; DSST: Digit symbol substitution test.

**Table 5 ijerph-20-05552-t005:** Univariate analyses to identify predictors of executive function in healthy controls (HC) and T2DM.

	HC *n* = 23		T2DM *n* = 19	
Executive Function	R	*p*	R	*p*
BMI (kg/m^2^)	−0.019	0.941	−0.274	0.305
Waist circumference (cm)	0.081	0.750	−0.274	0.304
Fat free mass (kg)	−0.305	0.218	−0.121	0.655
Fat mass (%)	0.528	0.024	0.049	0.857
Fasting glycemia (mmol/L)	0.053	0.840	0.447	0.083
triglycerides (mmol/L)	−0.172	0.510	0.219	0.415
Fasting insulin (pmol/L)	−0.377	0.317	−0.114	0.711
$\dot{V}$O_2_peak (mL/kg FFM/min)	−0.519	0.027	−0.300	0.260
VT1 (mL/kg FFM/min)	−0.524	0.026	−0.316	0.293
Circulatory Power (mL/min/kg/mmHg)	−0.534	0.023	−0.261	0.328
Cardiac Index (L/min/m^2^)	−0.466	0.051	−0.168	0.534
Left cardiac work index (kg/m/m^2^)	−0.305	0.219	−0.253	0.345
Systemic vascular resistance index (Dyn·s/cm^5^·m^2^)	0.370	0.131	0.081	0.766
tHb AUC effort	−0.486	0.041	−0.500	0.049
HHb AUC effort	0.456	0.057	−0.311	0.242
O_2_hb AUC effort	−0.620	0.006	−0.416	0.109
HbDiff AUC effort	−0.573	0.013	−0.205	0.447
tHb AUC recup	−0.265	0.287	−0.682	0.005
HHb AUC recup	0.514	0.029	0.055	0.846
O_2_hb AUC recup	−0.550	0.018	−0.686	0.005
HbDiff AUC recup	−0.650	0.004	−0.675	0.006

BMI: body mass index; VT1: 1st ventilator threshold; tHb: total hemoglobin; HHb: deoxyhemoglobin; O_2_Hb: oxyhemoglobin; AUC: area under the curve.

## Data Availability

All relevant data have been included in the paper. Due to ethical restrictions, individual data from the current study are available upon request from the corresponding author.

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
