# Peer review of "Acute Effects of a Maximal Cardiopulmonary Exercise Test on Cardiac Hemodynamic and Cerebrovascular Response and Their Relationship with Cognitive Performance in Individuals with Type 2 Diabetes"

_ijerph, 2023, doi:10.3390/ijerph20085552_

Round 1
Reviewer 1 Report
Abstract
Line 25: “Nineteen T2D patients”.
I suggest using person first language by changing “T2D patients” to “patients with T2D” or “people with T2D”. Please, check the entire manuscript.
Line 26: “…maximal cardiopulmonary exercise test…”
This term have already appeared with the abbreviation “CPET”.
Please, use the abbreviation in the entire text.
Line 28: “Prior to the maximal test”.
Please, use the term “CPET”.
Line 28: “the neuropsychological assessment”
In the title and background/objectives of the abstract, authors used “cognitive performances”. I suggest to keep using this term. It is not clear when authors use “the neuropsychological assessment” without no previous mention.
Line 30: “T2D patients”
Patients with T2D. Please, check the entire manuscript.
Line 31: “mL/min/kgFFM”
I have 3 concerns here. 1st: There is a typing mistake (“min” is the last unit in the equation). 2nd: I suggest using mL.kg-1.min.-1 instead of mL/kg/min. Please, consider that for the entire manuscript. 3rd. It is the first time “FFM” appears. The full term “fat free mass” should be mentioned.
Line 31: “Compared to HC, T2D patients showed lower peak cardiac index (p<0.05) and higher values of systemic vascular resistance index and systolic blood pressure at peak exercise (p=0.005). Cerebral HHb during the 1st and 2nd min of recovery was significantly higher in HC compared to T2D (p<0.05). Executive function performances (Z score) were significantly lower in T2D patients compared to HC (p=0.016).”
Please, consider offering some sort of magnitude of the differences. Once authors did not report the absolute values, reporting the % difference between groups would be of great value. Example: ““Compared to HC, T2D patients showed lower (xx %) peak cardiac index (p<0.05)”
Line 41: “peak exercise”
CPET.
Line 42: “with exercise”
CPET.
Keywords:
I suggest using only MeSH terms to improve the probability of finding your study in databases.
“NIRS” is not a MeSH term. Please, consider changing to “Spectroscopy, Near-Infrared” (there is a comma, it is not a mistake);
“type 2 diabetes” is not a MeSH term. Please, consider changing to “Diabetes Mellitus, Type 2”.
Lastly, although “exercise” is a MeSH term, it is more commonly used for conventional exercise routines (training). I suggest changing to “Exercise Test”. It would be improve the chances of your study to be detected in searches for systematic reviews regarding CPET.
Introduction
Line 47: “According to the International Diabetes Federation, 285 million people worldwide had diabetes in 2009”.
Data from 2009 are outdated. Please, offer some recent epidemiological data.
Line 60: “are decreased in individuals with T2D”.
How much decreased?
Line 63: “Regular physical activity”.
The abbreviation “PA” should be used here.
Line 85: In the Abstract, authors split the objectives into 3 points. Here, it was split into 2 points. I suggest authors to keep a standard. Choose one form and repeat in the entire manuscript.
Material and methods
Participants
Please, clarify how many participants were invited to participate, how many accepted/declined, how many were excluded, any data lost. I recommend the use of a flowchart diagram to clarify the recruiting process.
Also, clarify how these participants were invited / recruited.
Measurements
Line 113: “maximal cardiopulmonary exercise testing”
“CPET” was already used. It is not necessary to use the full term again.
Maximal cardiopulmonary exercise testing (CPET)
Line 120: “(depending on the patient’s fitness level)”. How was the fitness level determined? And what was the standard to determine the increment ratio?
Some questions:
1) Did both groups perform the same CPET protocol? According to your discussion (line 297), people with T2D can present with macro and microvascular dysfunction, muscle blood flow and mitochondrial dysfunction that could impact O2 extraction. Could not this limit the performance of a CPET (compared to healthy peers)? Should not a specific test for T2D be performed (maybe with lower increment ratio).
2) Throughout the manuscript authors attested that participants performed “a maximal CPET”. However, to determine the cardiorespiratory fitness, authors use the term “VO2peak”. The question is: was the CPET maximal? Conversely, in table 2 (results) authors use the term “VO2max”.
3) If it was maximal, why not using “VO2max” always?
4) Which criteria were used to determine the test was in fact maximal?
5) If the criteria were not attained, did authors exclude data? Or repeated the test?
6) Which criteria were used for test interruption?
There is much criticism around the use of VO2peak in research. Poole et al. for example, attest VO2peak is no longer acceptable (see doi:. 10.1152/japplphysiol.01063.2016).
In this sense, the use of criteria to determine if the VO2max was attained is necessary. The primary criteria for determining VO2max is the occurrence of a plateau in VO2. There are many thresholds being used to confirm the plateau, such as 2.1 mL.kg-1.min-1, 150 mL.min-1, 100 mL.min-1, 50 mL.min-1. More recently, the use of linear regression has also been adopted. Did authors analyze the VO2 plateau occurrence? If yes, which method / threshold was used.
As plateau is not always observed, some secondary criteria has been proposed (see Howley et al. 1995 PMID: 8531628) that include % heart rate max, respiratory exchange ratio, blood lactate concentration. These criteria, however, have also been criticized, as they can be satisfied at submaximal intensities, leading to false detection of VO2max. The use of a verification phase emerged as a possible tool to confirm the highest possible VO2 was attained in a CPET (see doi: 10.1139/h06-023, doi: 10.1371/journal.pone.0247057). The verification phase, in turn, is also not unanimity (doi: 10.3389/fphys.2018.00143). Which criteria did authors use to determine if the test was maximal (i.e. the VO2max was attained).
From the information provided, it is not possible to replicate your methods. Please, offer more detailed information on your CPET protocol.
Neuropsychological test battery
“Neuropsycological” or “cognitive performance”. Please, determine a stardard and use in the entire manuscript.
Line 169: “the Mini Mental state examination (MMSE)”
Please, insert citation.
Line 169: the Geriatric Depression Scale (GDS).
Please, insert citation.
Line 174: Digit Symbol Substitution Test (DSST).
Please, insert citation.
Line 176: Trail Making Test (TMT).
Please, insert citation.
Line 181: The D-KEFS Color-Word Interference Stroop Test.
Please, insert citation.
Line 192: e Rey Auditory Verbal Learning Test.
Please, insert citation.
Statistical analysis
Line 209: Data were summarized by mean ± standard deviation (SD). Normal Gaussian distribution of the data was verified graphically and by Shapiro–Wilk tests. Data were compared between the two groups using a Student t-test or a Mann-Whitney test.
Please, start the paragraph with the normality test. Then explain that groups were compared using a Student t-test or a Mann-Whitney test. However, it is important to say when each test was used (depending on the result of the test of normality). Lastly, attest that mean ± standard deviation (SD) were used to summarize data. Did author use mean ± standard deviation (SD) even when the data were not normally distributed?
Extra comments on methods
1) I missed a clear statement of how the recovery phase was conducted (time and body position.
2) I missed information about environmental condition. Did authors control air temperature and relative humidity? The absence of this control may impact on hemodynamic responses, leading to biased data.
3) I missed information about the calibration of the metabolic cart used to assess VO2max.
4) Did authors use facemask or mouthpiece during CPET?
5) Authors did not report how the ventilator threshold (reported in table 5) was determined.
6) There is no mention of verbal encouragement during CPET. Was that performed?
Results
Table 2
I missed the following results:
1) time to exhaustion.
2) Workload at ventilator threshold.
3) VO2 at ventilator threshold.
4) CPET increment ratio.
Figure 1
The term “PPO” in the graphics is not listed in the captions.
Discussion
Cardiopulmonary function and cardiac hemodynamic at peak exercise
Line 286: “We showed that T2D patients have reduced normalized ?̇O2peak (by FFM).”
Why do authors highlight the VO2peak by FFM (here and in the abstract)? It is not a common unit.
Line 286: “We showed that T2D patients have reduced normalized ?̇O2peak (by FFM) as compared to healthy controls, as well as reduced peak ventilation, indicating a reduced ventilatory convection. Pulmonary function (expiratory reserve volume and/or functional reserve capacity) can be impaired in obese patients (very prevalent in our T2D group)which increases respiratory load and cost [29].”
The group of people with T2D had 15.8% of smokers, while he HC had 0% (significantly difference; p = 0.048). Do not authors believe this could have influenced your results? The inclusion on smokers in only one group in a study focused on cardiorespiratory variables is a major limitation. I suggest removing this participants from analyses.
Cerebral hemodynamic during exercise and recovery
Line 301: “We showed that during exercise”.
Please, change “exercise” to CPET.
Line 307: “Kim et al. showed reduced PaCO2, cerebral blood flow and oxygenation”
How much reduction?
Line 310: “reduced 310 brain O2Hb, tHb and HHB during exercise compared to controls”
How much reduction?
Line 311: “We did not witness beneficial aerobic fitness effects on cerebral hemodynamics (O2Hb, tHb and HHB) in T2D in agreement with our previous study in obese patients [22]. Other studies showed that fitter young adults can have higher O2Hb, tHb and HHb values compared to less fit peers [21, 34].”
What could explain the differences between studies? Please, discuss it.
Line 324: “We also demonstrated that cerebral oxygen extraction can be improved with interval training (higher HHb levels) in obese patients [35].”
It is not relevant for the context of this manuscript, that should discuss the responses to CPET (a maximal test and not a submaximal exercise session).
Line 339: “We showed that individuals with T2D have reduced executive function”.
How much reduction?
Limitations and perspectives
The inclusion of smokers in only one (almost 16% of the group) is a major limitation that probably biased the results. My suggestion is to remove them from analyses.
The absence of criteria to determine the occurrence of VO2 plateau as well as the absence of a verification phase or employment of traditional criteria to confirm the VO2max are also important limitations. Please, include them in methos, otherwise as a limitation.
The absence of air temperature and humidity could also impact the hemodynamic results and should be mentioned.
Author Response
Reviewer 1
Comments and Suggestions for Authors
We would like to thank reviewer 1 for his comprehensive review of the manuscript. The suggestions will greatly improve the reading of the article. Please find below our responses to each question/comment.
Abstract
Line 25: “Nineteen T2D patients”.
Correction done
I suggest using person first language by changing “T2D patients” to “patients with T2D” or “people with T2D”. Please, check the entire manuscript.
Correction done
Line 26: “…maximal cardiopulmonary exercise test…”
Correction done
This term have already appeared with the abbreviation “CPET”.
Correction done
Please, use the abbreviation in the entire text.
Correction done
Line 28: “Prior to the maximal test”.
Correction done
Please, use the term “CPET”.
Correction done
Line 28: “the neuropsychological assessment”
Correction done
In the title and background/objectives of the abstract, authors used “cognitive performances”. I suggest to keep using this term. It is not clear when authors use “the neuropsychological assessment” without no previous mention.
Correction done
Line 30: “T2D patients”
Correction done
Patients with T2D. Please, check the entire manuscript.
Correction done
Line 31: “mL/min/kgFFM”
I have 3 concerns here. 1st: There is a typing mistake (“min” is the last unit in the equation). 2nd: I suggest using mL.kg-1.min.-1 instead of mL/kg/min. Please, consider that for the entire manuscript. 3rd. It is the first time “FFM” appears. The full term “fat free mass” should be mentioned.
For simplicity of VO2 normalization units, and because we published previously with slashes we kept them. Fat free mass was added.
Line 31: “Compared to HC, T2D patients showed lower peak cardiac index (p<0.05) and higher values of systemic vascular resistance index and systolic blood pressure at peak exercise (p=0.005). Cerebral HHb during the 1st and 2nd min of recovery was significantly higher in HC compared to T2D (p<0.05). Executive function performances (Z score) were significantly lower in T2D patients compared to HC (p=0.016).”
Please, consider offering some sort of magnitude of the differences. Once authors did not report the absolute values, reporting the % difference between groups would be of great value. Example: ““Compared to HC, T2D patients showed lower (xx %) peak cardiac index (p<0.05)”
We added some absolute values for means /SD in the abstract.
Line 41: “peak exercise”
Line 42: “with exercise”
We added CPET instead of “peak exercise” or “with exercise”
Keywords:
I suggest using only MeSH terms to improve the probability of finding your study in databases. “NIRS” is not a MeSH term. Please, consider changing to “Spectroscopy, Near-Infrared” (there is a comma, it is not a mistake); “type 2 diabetes” is not a MeSH term. Please, consider changing to “Diabetes Mellitus, Type 2”. Lastly, although “exercise” is a MeSH term, it is more commonly used for conventional exercise routines (training). I suggest changing to “Exercise Test”. It would be improve the chances of your study to be detected in searches for systematic reviews regarding CPET.
Thank you for these valuable details that will improve the visibility of the article. We have changed the keywords following your recommendations.
Introduction
Line 47: “According to the International Diabetes Federation, 285 million people worldwide had diabetes in 2009”.
Data from 2009 are outdated. Please, offer some recent epidemiological data.
We updated this number from the International Diabetes Federation internet site.
(https://diabetesatlas.org/)
Line 60: “are decreased in individuals with T2D”. How much decreased?
Thank you for your comment, we have clarified the sentence with the effects size. Thank you for your comment, we have clarified the sentence with the effects size. “More specifically, executive function [13], in particular cognitive flexibility and verbal fluency [14], and verbal episodic memory [14] are decreased in individuals with T2D with Cohen’s d ranged from -0.22 to -0.51 compared to healthy controls [12, 14, 15].”
Line 63: “Regular physical activity”.
Correction done
The abbreviation “PA” should be used here.
Line 85: In the Abstract, authors split the objectives into 3 points. Here, it was split into 2 points. I suggest authors to keep a standard. Choose one form and repeat in the entire manuscript.
We put the same 3 objectives points in the abstract.
Material and methods
Participants
Please, clarify how many participants were invited to participate, how many accepted/declined, how many were excluded, any data lost. I recommend the use of a flowchart diagram to clarify the recruiting process.
The COGNEX project is a large trial (not specific of diabete mellitus), we approached 455 participants for assessing eligibility, 318 participants were excluded for various reason (decline participation, not meeting inclusion criteria, incomplete data) and the final sample is 137 participants (with different CV profiles and diseases). This article focuses only on patients with diabetes.
Also, clarify how these participants were invited / recruited.
These participants were invited/recruited at the Montreal Heart Institute’s Preventive Medicine and Physical Activity Centre (ÉPIC) among our members principally via our medical teams and with posters.
Measurements
Line 113: “maximal cardiopulmonary exercise testing”
“CPET” was already used. It is not necessary to use the full term again.
Correction done
Maximal cardiopulmonary exercise testing (CPET)
Line 120: “(depending on the patient’s fitness level)”. How was the fitness level determined? And what was the standard to determine the increment ratio?
This section was rewritten and simplified. We refer to our previous published paper. The protocol increment was chosen according the subjects characteristic : age, biological sex, physical activity habitus (questioning the subject), and eventually a previous stress test (very frequent in our Centre ‘s members). This was done to choose (10-15 or 20 w/min) the optimal increment to have a time effort between 8 - 17 min. (M J Buchfurer 1983, Myer J 1992).
Some questions:
1) Did both groups perform the same CPET protocol? According to your discussion (line 297), people with T2D can present with macro and microvascular dysfunction, muscle blood flow and mitochondrial dysfunction that could impact O2 extraction. Could not this limit the performance of a CPET (compared to healthy peers)? Should not a specific test for T2D be performed (maybe with lower increment ratio).
There was not a specific protocol for HC or T2D, this was chosen on an individual basis. The power (watts) increment could be different from one person to another (10-15-20 W/min). See my comment above.
2) Throughout the manuscript authors attested that participants performed “a maximal CPET”. However, to determine the cardiorespiratory fitness, authors use the term “VO2peak”. The question is: was the CPET maximal? Conversely, in table 2 (results) authors use the term “VO2max”.
This has been changed in the document consistently for VO2max. We choose VO2max terminology for our healthy control group as well as the diabetic’s patients.
3) If it was maximal, why not using “VO2max” always?
Thank you for pointing out this mistake. Both the healthy control group and the diabetes patients were referred to using the VO2max nomenclature in all the manuscript.
4) Which criteria were used to determine the test was in fact maximal?
The main criterion was VO2 plateau (increase of VO2 values ≤150 ml/min during the last 30 secs). However, not all subjects met the VO2 plateau criteria (86% and 68% respectively for healthy control and T2D patients). The other two secondary criteria were exhaustion of the participant, not able to maintain the pace despite strong encouragement and a RER > 1.1 to have the best maximal effort for testing quality.
5) If the criteria were not attained, did authors exclude data? Or repeated the test?
All participants were pushed until exhaustion. All tests were performed by an experienced CPET examiner. We did not exclude any data and we did not repeat the test. We added some details in the methods regarding the end of the tests.
6) Which criteria were used for test interruption?
We added some precision in the methods: Criteria for maximal effort were the attainment of the primary maximal criteria: a levelling off of oxygen uptake (<150 mL/min) despite increased workload, or one of the three secondary maximal criteria: 1) a respiratory exchange ratio >1.10, 2) inability to maintain 60 rpm, 3) patient exhaustion due to fatigue or other clinical symptoms requiring exercise end (dyspnea, ECG and/or blood pressure abnormalities).
There is much criticism around the use of VO2peak in research. Poole et al. for example, attest VO2peak is no longer acceptable (see doi:. 10.1152/japplphysiol.01063.2016). In this sense, the use of criteria to determine if the VO2max was attained is necessary. The primary criteria for determining VO2max is the occurrence of a plateau in VO2. There are many thresholds being used to confirm the plateau, such as 2.1 mL.kg-1.min-1, 150 mL.min-1, 100 mL.min-1, 50 mL.min-1. More recently, the use of linear regression has also been adopted. Did authors analyze the VO2 plateau occurrence? If yes, which method / threshold was used.
We are aware of the discussion and debate in the literature regarding VO2max / VO2peak terminology. For that reason, we clearly defined that our VO2 values obtained are VO2max in our methods. We used maximal criteria oxygen uptake (<150 mL/min) and RER >1.10, however, not all the subjects reach these points. Briefly, 4 healthy control and 2 T2D were under RER < 1.1 (2 healthy participants have had a RER of 1.03 and 1.04; 1 T2D and 1 healthy participants both had an RER of 1.07, and 2 others have had a RER of 1.09. but they were no longer able to cycle despite strong encouragement. The VO2 plateau occurrence between the 2 groups was given in the CPET table : HC : 86 %, T2DM : 68 %.
As plateau is not always observed, some secondary criteria has been proposed (see Howley et al. 1995 PMID: 8531628) that include % heart rate max, respiratory exchange ratio, blood lactate concentration. These criteria, however, have also been criticized, as they can be satisfied at submaximal intensities, leading to false detection of VO2max.
We fully agree with this point. This is why the VO2 plateau or subject exhaustion should be the main criteria. We have clarified in our CPET methods these aspects : 1) a respiratory exchange ratio >1.10, 2) inability to maintain 60 rpm, 3) patient exhaustion due to fatigue or other clinical symptoms requiring exercise end (dyspnea, ECG and/or blood pressure abnormalities). We did not us % of max heart rate and/or blood lactate (not measured).
The use of a verification phase emerged as a possible tool to confirm the highest possible VO2 was attained in a CPET (see doi: 10.1139/h06-023, doi: 10.1371/journal.pone.0247057). The verification phase, in turn, is also not unanimity (doi: 10.3389/fphys.2018.00143). Which criteria did authors use to determine if the test was maximal (i.e. the VO2max was attained).
To respond to that point, the verification phase testing has been a big debate recently. We did not performed this test in our study, there was not such consideration at that time. This test might be particularly difficult in some clinical populations, and is really not performed in clinical context research. This could be pertinent in interventional exercise training studies, as also the technical error or reproducibility of the CPET procedure. We believe that for cross sectional comparison study like ours, this aspects maybe less important. Moreover, recent meta-analysis suggest that VO2max achieved during the 2 procedure are very close (PMID: 33596256). All our tests were performed until exhaustion and/or VO2 plateau.
From the information provided, it is not possible to replicate your methods. Please, offer more detailed information on your CPET protocol.
Clarification were done in the methods regarding the CPET methods and protocol. I added some details here, and the section had to be re-written for autoplagiarism modifications asked by the journal. We think that our methods is easily replicable, as we have published it in many work previously in various clinical populations, including obese patients, and cardiac patients. Our methods used in the CPET is very common of what is used for CPET in clinical populations in research.
Neuropsychological test battery
“Neuropsycological” or “cognitive performance”. Please, determine a stardard and use in the entire manuscript.
Line 169: “the Mini Mental state examination (MMSE)”
Line 169: the Geriatric Depression Scale (GDS).
Line 174: Digit Symbol Substitution Test (DSST).
Line 176: Trail Making Test (TMT).
Line 181: The D-KEFS Color-Word Interference Stroop Test.
Line 192: e Rey Auditory Verbal Learning Test.
Please, insert citation.
We have added the 6 requested references on neurospychological tests.
Statistical analysis
Line 209: Data were summarized by mean ± standard deviation (SD). Normal Gaussian distribution of the data was verified graphically and by Shapiro–Wilk tests. Data were compared between the two groups using a Student t-test or a Mann-Whitney test. Please, start the paragraph with the normality test. Then explain that groups were compared using a Student t-test or a Mann-Whitney test. However, it is important to say when each test was used (depending on the result of the test of normality). Lastly, attest that mean ± standard deviation (SD) were used to summarize data.
We have modified the ''statistics'' paragraph
Did author use mean ± standard deviation (SD) even when the data were not normally distributed?
Yes.
Extra comments on methods
1) I missed a clear statement of how the recovery phase was conducted (time and body position.
This was defined in the methods (CPET and NIRS): 5 min of recovery, seated on the bike, 2 min active recovery pedaling and 3 min passive recovery.
2) I missed information about environmental condition. Did authors control air temperature and relative humidity? The absence of this control may impact on hemodynamic responses, leading to biased data.
This information was added in NIRS section: The room temperature and relative humidity were constant in the laboratory (21 °C, 25 %).
3) I missed information about the calibration of the metabolic cart used to assess VO2max.
We added this infos. In the methods. The calibration of the flow module was accomplished by introducing a calibrated volume of air at several flow rates with a 3-liter pump. Gas analyzers were calibrated before each test using a standard certified commercial gas preparation (O2: 16%; CO2 : 5%)
CO2: 5%).4) Did authors use facemask or mouthpiece during CPET?
We used a facemask, we added this info. in the methods.
5) Authors did not report how the ventilator threshold (reported in table 5) was determined.
The information was added in the CPET methods : The 1st ventilatory threshold (VT1) was determined using a combination of the V-slope, ventilatory equivalents, and end-tidal oxygen pressure methods.
6) There is no mention of verbal encouragement during CPET. Was that performed?
Yes, this was added in the CPET methods.
Results
Table 2
I missed the following results:
1) time to exhaustion.
2) Workload at ventilator threshold.
3) VO2 at ventilator threshold.
4) CPET increment ratio.
These data were added in the table 2, with their stats.
Figure 1
The term “PPO” in the graphics is not listed in the captions.
The abbrev. definition was done in the captions.
Discussion
Cardiopulmonary function and cardiac hemodynamic at peak exercise
Line 286: “We showed that T2D patients have reduced normalized ?̇O2peak (by FFM).”
Why do authors highlight the VO2peak by FFM (here and in the abstract)? It is not a common unit.
Due to difference in body composition (higher fat mass in T2DM), the normalization with FFM of VO2 is more accurate to allow group comparison and is less biases by these confounding factors. We and other have used it in the past in obese subjects to allow comparisons with non-obese subjects. We agree that this is not the most common unit.
Line 286: “We showed that T2D patients have reduced normalized ?̇O2peak (by FFM) as compared to healthy controls, as well as reduced peak ventilation, indicating a reduced ventilatory convection. Pulmonary function (expiratory reserve volume and/or functional reserve capacity) can be impaired in obese patients (very prevalent in our T2D group) which increases respiratory load and cost [29].”
The group of people with T2D had 15.8% of smokers, while he HC had 0% (significantly difference; p = 0.048). Do not authors believe this could have influenced your results? The inclusion on smokers in only one group in a study focused on cardiorespiratory variables is a major limitation. I suggest removing this participants from analyses.
We agree that the presence of cardiovascular risk factors (CVRF) (other than diabetes) such as smoking, hypertension, hyperlipidemia, etc. can modulate cerebral arterial vasomotricity. However, clinically, a type 2 diabetic patient always has other CVRF, and it is impossible to recruit only T2D patients without other CVRF. For this, we want to keep these 3 patients. Similar results were found for VO2peak, Ventilation and RER after exclusion of these 3 smokers.
Cerebral hemodynamic during exercise and recovery
Line 301: “We showed that during exercise”.
Please, change “exercise” to CPET.
Correction done.
Line 307: “Kim et al. showed reduced PaCO2, cerebral blood flow and oxygenation”
How much reduction?
We added in the discussion line : PaCo2 : -0.8 mmHg, CBF : 25 % less appro., ScapO2% ; 2.5% less,
Line 310: “reduced 310 brain O2Hb, tHb and HHB during exercise compared to controls”
How much reduction?
Information added in the discussion (-1.25; -6; -6.5 µM at 90-100% O2max)
Line 311: “We did not witness beneficial aerobic fitness effects on cerebral hemodynamics (O2Hb, tHb and HHB) in T2D in agreement with our previous study in obese patients [22]. Other studies showed that fitter young adults can have higher O2Hb, tHb and HHb values compared to less fit peers [21, 34].”
What could explain the differences between studies? Please, discuss it.
We have modified this part to better reflect the complexity between VO2max and brain circulation, please see PMID: 36074924 : We added the sentence in the paragraph : “Age, gender, genetic factors, insulin resistance, or others cardiovascular risk factors and perhaps drug treatments are all factors that can influence arterial vasomotricity and the relationship between cardiorespiratory fitness and cerebral circulation, PMID: 36074924”.
Line 324: “We also demonstrated that cerebral oxygen extraction can be improved with interval training (higher HHb levels) in obese patients [35].”
It is not relevant for the context of this manuscript, that should discuss the responses to CPET (a maximal test and not a submaximal exercise session).
We are mentioning one of our previous study realized in obese subjects during peak effort and recovery. Brain NIRS were measured according to the same methods. The only difference was it was a pre-post training measure, with higher brain HHB in post training. A higher HHB was mentioned to reflect a higher O2 extraction by the tissue. In that line, we were trying to explain HHB difference between our 2 groups in the present study, with other potential mechanisms.
Line 339: “We showed that individuals with T2D have reduced executive function”.
How much reduction?
Cognitive Z-score on executive function: -0.40 vs 0.14 respectively for T2D and HC (p=0.016) (please see the table 4, on the results section).
Limitations and perspectives
The inclusion of smokers in only one (almost 16% of the group) is a major limitation that probably biased the results. My suggestion is to remove them from analyses.
We agree that the presence of cardiovascular risk factors (CVRF) (other than diabetes) such as smoking, hypertension, hyperlipidemia, etc. can modulate cerebral arterial vasomotricity. However, clinically, a type 2 diabetic patient always has other CVRF, and it is impossible to recruit only T2D patients without other CVRF. For this, we want to keep these 3 patients. Similar results were found for VO2peak, Ventilation and RER after exclusion of these 3 smokers.
The absence of criteria to determine the occurrence of VO2 plateau as well as the absence of a verification phase or employment of traditional criteria to confirm the VO2max are also important limitations. Please, include them in methos, otherwise as a limitation.
These points have been addressed previously in my responses to your comments and in the methods section.
The absence of air temperature and humidity could also impact the hemodynamic results and should be mentioned.
This points has been addressed previously in my responses to your comment and in the methods.

Reviewer 2 Report
The present study aims to compare cognitive performances, cerebral oxygenation/perfusion (oxy, deoxy and total hemoglobin: O2Hb, HHb and tHb) and cardiac hemodynamic during a CPET and its post-exercise period (recovery) in T2D 19 patients vs. 22 healthy controls. And to examine if V̇O2 peak, cardiac output and cerebral oxygenation/perfusion are related to cognitive performances.
Abstract
Line 30 and 31 – Please, correct the V̇O2 description and the unity. (mL·kgFFM-1·min-1)
Introduction
Lines 85-89 – The final of introduction shows 3 objectives and at abstract have 2 objectives. Please keep the standart.
Figure 1A and 1B – Did the author´s use t test for compare each moment for inferential statistics? I believe that was used ANOVA with Bonferroni post hoc test. Please report this sentence at statistics and if the all criteria was attended (Normality, Variance…)
Table 5 – The author´s use VT1 – Please include the description for determining the first ventilatory threshold in the methods. Corrected "ventilator" and insert ventilatory at description.
Author Response
We would like to thank reviewer 2 for his comprehensive review of the manuscript. The suggestions will greatly improve the reading of the article. Please find below our responses to each question/comment.
Reviewer 2
The present study aims to compare cognitive performances, cerebral oxygenation/perfusion (oxy, deoxy and total hemoglobin: O2Hb, HHb and tHb) and cardiac hemodynamic during a CPET and its post-exercise period (recovery) in T2D 19 patients vs. 22 healthy controls. And to examine if V̇O2 peak, cardiac output and cerebral oxygenation/perfusion are related to cognitive performances.
Abstract
Line 30 and 31 – Please, correct the V̇O2 description and the unity. (mL·kgFFM-1·min-1)
We have used the / use instead of the -1, for better clarity and we previously published with that format in the same journal.
Introduction
Lines 85-89 – The final of introduction shows 3 objectives and at abstract have 2 objectives. Please keep the standart.
Thank you for pointing out this mistake, we put the same 3 objectives points in the abstract.
Figure 1A and 1B – Did the author´s use t test for compare each moment for inferential statistics? I believe that was used ANOVA with Bonferroni post hoc test. Please report this sentence at statistics and if the all criteria was attended (Normality, Variance…)
First, normality of data was verified for each time point with a Shapiro wilk test. Then, we have used a t test (normal distribution) or a Man Whitney (non para for abnormal dis.) for each time point to compare the groups.
Table 5 – The author´s use VT1 – Please include the description for determining the first ventilatory threshold in the methods. Corrected "ventilator" and insert ventilatory at description.
The VT1 determination was added in the methods (CPET section): The 1st ventilatory threshold (VT1) was determined using a combination of the V-slope, ventilatory equivalents, and end-tidal oxygen pressure methods.

Reviewer 3 Report
1. Nineteen individuals is small number of patients, did the authors perform power calculation? How did the investigators know that this is not lack of power?
2. How did the investigators select those 19 patients? any selection bias?
3. Hemoglobin also affects A1C, Hb levels should be reported/
4. Heigh is misspelled in the Table 1
5. Footnotes and abbreviations should be provided for all tables and figures.
6. Medications that used for T2 diabetes and also antihypertensive medications such as Beta Blocker may affect exercise capacity.
Author Response
We would like to thank reviewer 3 for his comprehensive review of the manuscript. The suggestions will greatly improve the reading of the article. Please find below our responses to each question/comment.
Reviewer 3
Comments and Suggestions for Authors
- Nineteen individuals is small number of patients, did the authors perform power calculation? How did the investigators know that this is not lack of power?
We agree with the reviewer that we have a small number of T2DM. We did not perform a power calculation.
- How did the investigators select those 19 patients? any selection bias?
Patients were selected among our members via our medical team and via posters in our Center and hospital. We can not exclude a selection bias because of recruitment in one Center.
- Hemoglobin also affects A1C, Hb levels should be reported.
We do not have these data unfortunately.
- Heigh is misspelled in the Table 1
The correction was made.
- Footnotes and abbreviations should be provided for all tables and figures.
The correction was made.
- Medications that used for T2 diabetes and also antihypertensive medications such as Beta Blocker may affect exercise capacity.
This was added in the table 1 and in the limitation.
